# Sociocultural Influences on Dietary Practices and Physical Activity Behaviors of Rural Adolescents—A Qualitative Exploration

**DOI:** 10.3390/nu11122916

**Published:** 2019-12-02

**Authors:** Mohammad Redwanul Islam, Jill Trenholm, Anisur Rahman, Jesmin Pervin, Eva-Charlotte Ekström, Syed Moshfiqur Rahman

**Affiliations:** 1Department of Women’s and Children’s Health, Uppsala University, SE-751 85 Uppsala, Sweden; mohammadredwanul.islam@kbh.uu.se (M.R.I.); jill.trenholm@kbh.uu.se (J.T.); lotta.ekstrom@kbh.uu.se (E.-C.E.); 2International Center for Diarrheal Disease Research, Bangladesh (ICDDR,B), Dhaka 1212, Bangladesh; arahman@icddrb.org (A.R.); jpervin@icddrb.org (J.P.)

**Keywords:** dietary practices, adolescent, physical activity, Bangladesh

## Abstract

In the aftermath of nutrition transition and ever-increasing sedentarism, adolescents globally are exposed to negative health consequences. Diverse sociocultural influences play a critical role in their adoption of unhealthy dietary practices and suboptimal physical activity behaviors. Context-specific understandings of how these sociocultural influences shape adolescents’ dietary and physical activity patterns in a rural, resource-limited setting remained elusive. Aiming to address the gap, this qualitative study explored adolescents’ and mothers’ perception of broader sociocultural aspects that sculpt the food choices, eating habits and physical activity behaviors of adolescents in Matlab, Bangladesh. Six digitally-recorded focus group discussions were transcribed verbatim, translated into English and analyzed thematically. Marked taste-driven dietary preference of adolescents and its prioritization within family by the mothers, popularity of street foods, better understanding of the importance of food hygiene and safety contrasting with narrow perception of balance and diversity in diet, peer influence along with deficient school and community food environment, internalization and rigidity of gender norms were found to be exerting major influence. The findings highlighted key targets for community-based nutrition interventions and endorsed thorough consideration of socio-cultural factors in formulating strategies to promote healthful eating and physical activity behaviors among the adolescents.

## 1. Introduction

Adolescence is a transitional life phase characterized by marked biopsychosocial changes potentially capable of determining the future health status in adulthood. It is also during this phase that habits are established and health-related behaviors, either positive or negative, are adopted, often persisting throughout adulthood [1]. The outcomes of demographic and nutritional transitions in the preceding decades with simultaneous socioeconomic changes have put adolescents at the core of many burning global and public health challenges [2]. The transition from widespread undernutrition to increasing overweight and obesity has emerged as a complex challenge facing the adolescents in low- and middle-income countries (LMICs) historically affected by food insecurity. The food environment in LMICs changed dramatically as globalization of food markets, lowering of manufacturing cost of edible oils and caloric sweeteners, and increased production of animal-source foods [3] paved the way for abundant availability of cheap, energy-dense foods and beverages. Although adolescents’ energy intake has spiralled, the diets remain limited in diversity [4], leading to dietary patterns unconducive to a healthy lifespan free from nutrition-related non-communicable diseases (NCDs). Unhealthy dietary practices and eating habits among adolescents involving less consumption of fruits and vegetables [5,6,7], snacking [8], frequent consumption of sugar-sweetened beverages (SSB) [5,9] and tendency to skip breakfast [6,7,10,11,12] combined with an increase in screen-based recreational behaviors like watching television, playing games and using social media [5] have exposed adolescents globally to a higher likelihood of adverse health outcomes in their ensuing adulthood. Exploration of dietary and physical activity behaviors in this population is, therefore, of notable public health interest.

Furthermore, sociocultural and sociodemographic factors have been shown to play a crucial role in sculpting dietary patterns of the adolescents [13,14,15,16]. One interesting revelation from the 2016 Healthy Active Kids South Africa (HAKSA) report was significantly lower participation of adolescent girls in the health-enhancing moderate-to-vigorous physical activity (MVPA) [17]. Similarly, it has been reported that adolescent females experienced more sociocultural barriers to following a healthy physical activity pattern than adolescent males in seven Arab countries [18]. These findings indicate a major influence of sociocultural and gender issues on what adolescents eat and how physically active they are. However, context-specific influences of sociocultural factors on adolescents’ dietary patterns and physical activity behaviors, particularly in rural settings of LMICs, remained less explored.

While the majority of qualitative studies on adolescents’ diet and physical activity have been conducted in urban settings with higher standards of living, this study aimed to explore adolescents’ dietary practices, theirs and their mothers´ perceptions of healthy eating, patterns of their physical activities, their understanding of a healthy adolescent physique and how various sociocultural influences shape these in the context of rural Bangladesh.

### Conceptual Framework

The life course perspective offers a theoretical scheme to guide studies on adolescence and adolescent health issues. It posits that no life phase can be studied and conceptualized in isolation, insisting that an improved understanding of adolescence, and thus factors affecting adolescent health, requires locating and viewing this unique phase within the larger context of the full life course [19,20]. Therefore, acknowledgement of adolescents’ health status as one resulting from critical interaction between the prenatal and early childhood development and the social determinants of health which affect the adoption of health-related behaviors [2] is pivotal to research on such important avenues of adolescent health as eating habits, food preferences, physical activity pattern and so forth. Unsurprisingly, many of these determinants remain largely outside the fundamental scope of health services [21]. Developing on this theoretical foundation we perceive social experiences pertinent to adolescence, sociocultural influences, and gender aspects as important modifiers of adolescents’ dietary and physical activity behaviors. The exploratory and interpretative pursuit stemmed from our assumption that adolescents’ health-related behaviors are sculpted by the opportunities and constraints that they come across under a particular sociocultural circumstance. Moreover, the analysis of qualitative data was guided by social constructionism [22] in an attempt to shed light upon how adolescents embrace certain dietary and physical activity patterns against the backdrop of rural sociocultural landscape in a LMIC. Viewing through the lens of social constructionism enabled us to approach the adolescents’ own account of their reality regarding diet and physical activity in a rural, collectivistic society with a gendered perspective.

## 2. Materials and Methods

Our study availed the opportunity presented by the qualitative arm of 15 year follow up of the Maternal and Infant Nutrition Interventions in Matlab trial (MINIMat, ClinicalTrials.gov identifier ISRCTN16581394), which was a randomized supplementation trial initiated in Matlab in 2001 to study the impact of food and micronutrient supplementation on pregnancy outcomes [23]. Matlab, located 55 km to the southeast of the capital Dhaka, is a riverine rural sub-district in Chandpur and has been the site of the Health and Demographic Surveillance System (HDSS) run by the International Centre for Diarrhoeal Disease Research, Bangladesh (icddr,b) since 1966 [24]. By the end of 2003, a total of 4436 pregnant women were enrolled in the trial, resulting in 3625 live births [23]. These offspring have been repeatedly followed up, including one follow-up at 15 years of age, that has been recently completed. Qualitative data for this study were collected as a part of this latest follow-up.

### 2.1. Participants

Six focus group discussions (FGDs) were conducted in 2017 with 6–8 participants per group. Out of the six, two focus groups consisted of mothers of the adolescents, two of adolescent boys only, and two of adolescent girls only. Participants were recruited purposively and comprised adolescents living in the study area since birth and their mothers. The Matlab HDSS keeps track of socio-demographic events through household visits by community health workers (CHWs) every two-month. The CHWs invited adolescents and their mothers for participating in the study. Most, but not all, of the participating adolescents belong to the MINIMat cohort. Although a growing sense of autonomy marks adolescence [25], in the rural, agrarian communities of Bangladesh they live with parents. Their mothers are the ones who carry out household responsibilities of cooking as well as deciding on food-related purchases and what to cook for and feed their children. Mothers could also affect adolescents´ dietary practices through specific parenting styles and expression of normative beliefs [26]. Therefore, the role of mothers was deemed crucial considering the topic under study in that particular setting. Semi-structured discussions in Bengali were facilitated by a trained, female field research assistant, in the presence of an investigator (SMR). The FGDs took place in the icddr,b Matlab premise and lasted from 60 to 85 min. Some of the participants knew each other beforehand, as schoolmates or neighbors. The adolescents were 14–17 years old and the mothers were in their thirties and forties. Regarding educational status, the adolescents had a minimum of primary education and some of the mothers were illiterate.

### 2.2. Data Collection

The focus groups addressed the following broad lines of inquiry:Foods, snacks and drinks commonly consumed by adolescents in Matlab;Any gender-based difference in dietary preferences and physical activity behaviors;Link between foods, physical activity and health;Adolescents’ idea about a healthy and desirable physique or body size;Physical activities and sports that the teenagers in Matlab engage with;Adolescents’ perception of health benefits from physical activities.

The focus of the discussions was to capture participants’ accounts of eating habits, food choices and physical activity and to elicit their perspectives on how the sociocultural milieu influenced the gendered dimensions, that, among other factors, catalyze these determinants of adolescent health. Elucidation of the variation in informants’ construction of reality surrounding the topic received utmost attention during the FGDs. Necessary probing was done to better understand the nuances in their perceptions. Adolescents were encouraged to speak in their own words and to clarify individual and shared outlooks. The FGDs were audio recorded, transcribed verbatim in Bengali and translated into English by the first author and checked for accuracy by the last author, as both are native Bengali speakers.

### 2.3. Data Analysis

Analysis of the textual data followed the process of inductive thematic analysis put forward by Braun and Clarke [27]. As thematic analysis offers a systematic yet flexible approach toward identification and analysis of patterns of meaning in the data, it suited the purpose of this paper best. Transcripts were repeatedly read, and various aspects discussed among co-authors; discussions often involved Socratic questioning [28] within the research team, consisting of both insiders to the Bangladesh context and outsiders less familiar to that, thereby challenging any taken-for-granted notions expressed by the insiders. The diverse research team included academic professionals from qualitative and quantitative domains. The analysis was data-driven and interpretive, leading to the identification of relevant data extracts, followed by a manual coding process that went beyond the explicit meaning to capture preconceptions and assumptions underlying what the informants had said or expressed. The codes were then organized into themes that encompass meaningful patterns and repeated resemblances or differences of significance. Table 1 exemplifies the coding process. The analytical pursuit was dynamic and non-linear with researchers navigating to and fro across the transcripts to ensure that the themes convey the essence of the data.

### 2.4. Ethics Statement

This study was framed within the research project of 15-year follow-up of the MINIMat trial that received approval by the Ethical Review Committee at icddr,b in Dhaka, Bangladesh. Written informed consent and assent were obtained from the participating mothers and adolescents respectively following full disclosure and detail explanation of the purpose and methods as well as the confidential, anonymous handling of personal information and collected data.

## 3. Results

Figure 1 shows the four major themes that emerged from the analysis. Key sub-themes of crucial relevance to the overall research aim are listed under the themes. These sub-themes will structure the presentation of the findings that follows. The horizontal axis on the top of the figure focuses on adolescence within the broader life course and conceptually localizes our findings at the interface between adolescents’ dietary and physical activity behaviors, and socio-cultural influences shaping that.

### 3.1. Adolescents’ Taste-Driven Food Choice Reigns Supreme

#### 3.1.1. Mothers Succumbing to Adolescents’ Food Choices

The participant mothers claimed that the adolescents have a dislike of vegetable and fish, and a remarkably taste-driven food preference as well. Corroborating that, the adolescents acknowledged their food choices to be largely shaped by their liking for savory foods that “taste better”. On one hand, this led adolescents to frequently turn to street foods with a tendency to avoid vegetable and fish, in spite of knowing the health benefits offered by the latter. On the other hand, mothers invariably tried to comply with their children’s food choices, and thus, were often compelled to cook a separate item or dish for the adolescent member/s of the family:

*“If they are asked to eat vegetables, they would deny and ask for meat... We (the parents and other older members of the family) find broth prepared with papaya or ash gourd* (green pumpkin) *tasty, but they would never like that. They like bhuna* (a very oily curry cooked by oil-roasting spices and later adding water as a moistening agent)*-whether its boiled egg bhuna or meat bhuna. We adults don’t mind consuming vegetables.”*(Mother, FGD 1)


*“Although I know vegetables keep the body healthy and boost our blood, I don’t eat them...... I don’t like them at all, I just can’t eat vegetables.”*
(Adolescent girl, FGD 1)


*“Although the outside foods are worse than those prepared at home, they taste better and there are food items that can’t be made at home.”*
(Adolescent boy, FGD 1)

Mothers revealed that without the family meal containing at least one dish that meets the taste-driven preference of the adolescent member/s of the family, often, the adolescents refuse to eat, ending up skipping the meal or opting for street food instead, which mothers consider to be unhealthy. In line with that, the adolescents spoke frankly about their appreciation of the “better taste” of street foods which they knew were substandard to home-made foods in many aspects.

#### 3.1.2. Snacks and Street Foods Attract Adolescents More

Snacking and regular consumption of street foods consumption were noteworthy dietary practices among adolescents in Matlab. They consume a number of foods as snacks, such as puffed rice, oil-stirred black chickpeas, chanachur (a snack mix containing dried and spiced ingredients like fried lentils, fried peanuts and chickpeas, fried onions, flaked rice), instant noodles, shingara, samosa, lentil fritters, potato chips, biscuit with tea, packaged cake or muffin and various street foods. Fruit and salad were not mentioned as a common snack. Occasionally, adolescents have roti (round flatbread made from whole-wheat flour known as atta) in the breakfast. Mothers opined that instant noodles are better alternative to the street foods, and they prepare that for the adolescents when they snack at home expecting health benefits from that.

*“For me, nothing is as good as foods prepared at home. Outside foods are always bad. But do the kids understand this! If I try to make her* (participant’s daughter) *understand today, the very next day, she would eat those in the school. I advise her not to eat those outside foods. She has a problem with dyspepsia. I tell her [that] I would make you noodles at home, you could eat that before going to school and take that as tiffin with you. Even then she asks for 10–15 taka to buy jhalmuri* (an oily mix of puffed rice, green chili, sliced onion, mustard oil, and an assortment of spices), *which is also not good, very oily.”*(Mother, FGD 1)


*“My eating pattern is kind of similar to him. Usually, it’s either roti or rice in the morning, I eat rice three times a day. In between the main meals I have snacks, and these are usually oil-fried foods... [like] shingara, samosa, lentil fritters...... these we buy in the afternoon when we go out to roam around.”*
(Adolescent boy, FGD 2)

Snacking commonly occurs in the afternoon when there is a break given in the school, after school on the way back to home, and at home in the evening when adolescents start studying. Adolescents buy and eat foods sold by street vendors on most days of the week and popularly consumed are fuchka (known as pani puri in parts of India), chotpoti (boiled white chickpeas served in a sour soup along with boiled diced potato, sliced onion and chili pepper, different spice powders, and grated boiled egg on top), jhalmuri, and puri (a deep-fried bread with spiced red lentil or mashed potato inside). Adolescents are also found to be fond of sugar-sweetened beverages. While the adolescents acknowledged that their food choice is different from that of adult and senior members of their families, and that those family members don’t consume street foods as much as they do, they linked this to health problems experienced by the adults.


*“Because of gastric (local layman expression for dyspepsia, abdominal bloating, fullness, belching and discomfort) many of the adult people don’t eat oil-fried foods; but we love those very much; so, there are differences in food choice.”*
(Adolescent boy, FGD 2)

#### 3.1.3. Seasonal Variation in Access to Home-Grown Fruits

The mothers narrated their satisfaction regarding the opportunity for their adolescent children to enjoy the goodness of a variety of summer fruits, as families commonly own fruit trees surrounding the house yards that yield mango, jackfruit, papaya, guava, lemon, litchi, pomegranate and so forth. One of the mothers also said that this opportunity of rural adolescents to enjoy fresh fruits directly from the trees is something unknown to adolescents living in the cities. The perceived importance of fruit consumption by adolescents is reflected by the household tradition of not selling these fruits.


*“The fruits that are from the trees at home-mango, litchi-we can’t buy [these from the market], but the kids can eat from the trees. It’s not always possible for us to buy fruit. But we don’t sell the fruits grown at home, we keep that for the children, we tell them to eat as much as they can, also there is guava. It’s now the season for mango, they even eat ten mangoes a day!”*
(Mother, FGD 1)


*“Now this is the season for mango and [other] summer fruits here, we eat fruits quite a lot during this time of the year. After the season, we get to eat fruits rarely; … these need to be bought from the market and often are costly.”*
(Adolescent girl, FGD 2)

While mothers expressed their satisfaction about this contextual advantage of rural residence, they recognized and reasoned how fruits are rarely consumed by the adolescents for the rest of the year because of the higher retail price of fruits. Beyond the summer and early monsoon, consumption of fruit declines to a level that the adolescents do not have “at least one fruit a day”, and again, this was linked, by the mothers and adolescents alike, to lack of affordability.

### 3.2. High Knowledge of Hygiene and Food Safety Not Evident in Adolescents’ Actual Dietary Practice

The mothers as well as the adolescents highly valued the hygienic preparation of foods and the food safety aspect in terms of foods and fruits being free from preservatives, pesticides and chemicals. Mothers labelled preservatives and chemicals sprayed over fruits as “medicines” and made sense of it by interpreting presence of these “medicines” in fruits bought from market as a threat to health. Participants believed lack of hygienic preparation to be the most important factor rendering foods prepared and sold by vendors in the streets inferior to foods cooked at home. Mothers claimed that they attempt to pay relentless attention to basic hygiene of food preparation when they cook food at home and conveyed that this largely makes the home-made foods healthier. Informants viewed foods bought and consumed outside as persistently exposed to microbes, because of not being covered and handled “properly”. They also noted the tendency of street food vendors to not wash hands regularly.


*“These [foods] are sold by the vendors in the market, not even covered properly; so, there must be dirt and dust, germs, and god knows what else! They cut the ingredients without washing, [they] don’t even wash their hands before preparing those foods.”*
(Adolescent girl, FGD 2)

Mothers voiced their apprehension of the presence of various chemicals in the fruits available in the market and discussed that they frequently come across this issue through news broadcasted in electronic media. The participants took notice of incidences where eating outside foods presumably led to upset stomach and indigestion, and considered that eating foods on a regular basis from restaurants or street vendors could be detrimental to health. However, such extensive awareness about hygiene and food safety neither translated into strict avoidance of street foods by the adolescents nor stopped mothers from buying their children’s favorite fruits from the market. Adolescents’ inclination toward street foods apparently was unaffected by prevailing impression of low hygienic standard of the street foods.


*“We often get to watch and listen to news about this issue (chemicals in fruits) in the television. What can we do? Many of the fruits are not grown in our locality, for example, mango grows here in our yards and we don’t have to buy mango from market. But, when it comes to litchi, a fruit that the kids like, we buy it from market, even though we know it contains medicine and that we give to the kids and eat ourselves.”*
(Mother, FGD 2)


*“Foods are prepared at home with care. The outside foods are prepared without care or attention to hygiene … I eat [those] but not frequently. What can we do! We just have to [eat outside foods]!”*
(Adolescent boy, FGD 1)

According to the participants, the essential quality of good food is the ability to provide sufficient energy to keep the body strong and diet needs to be sufficiently energy-dense to promote health. Mothers, who usually take care of household cooking and food preparation, expressed their concern over the prices of “healthy foods” claiming that their inability to afford hinders regular consumption of these foods by the adolescents. Nevertheless, it was also argued that a broader knowledge of how to capitalize on locally ubiquitous vegetables, such as kochu (taro corms, edible root of *Colocasia esculenta*) or shaak (common Bengali name for leafy vegetables like red amaranth, spinach, et cetera) might help in diversifying the diet and making it rich in nutrients.


*“... egg, milk, meat, fruits like apple, orange... If we can get these for the kids, they will get nutrition, they will have energy in their body, the brain will work better. But, you see, we need to have money for these. So, the kids actually eat whatever we can afford.”*
(Mother, FGD 2)


*“Shaak, however, is nutritious; although many people use to think this is just ordinary shaak. But it contains vitamins and vitamins are good. To me, shaak is even better than meat. Here in the villages, if you just go out or just look across the yard you can get shaak and let the kids eat that, although we don’t do this out of laziness. We think it would do good, only if we can give them meat and fish.”*
(Mother, FGD 2)

### 3.3. Peer Influence in the Context of a Poor-Quality Food Environment Dictating Adolescents’ Food Choices

Out of a number of contextual factors touched upon by the informants in the FGDs, peer influence emerged as an aspect repeatedly talked about. As the adolescents reported spending a significant part of the day with their peers, their dietary practices also appeared to be converging on a pattern established largely on a collective preference of street foods that they enjoy consuming together as well as sharing with each other.

#### 3.3.1. Bonding over Street Food

Street food was frequently mentioned, both by adolescent girls and boys, as something that brings peers together by creating an aura of tightly-knit companionship, and thus molds how they socialize. The adolescents perceived the street foods as “part and parcel” of the time they spend together. Street foods were claimed to taste even better when consumed with a group of friends or classmates, as they mingle with peers over these street foods, and often if not always, share these foods with each other.


*“It is different when we are with our friends and we have lot of fun eating these together, buying from the vendors. It actually is a part of our time that we spend together.”*
(Adolescent girl, FGD 1)


*“After finishing the school I don’t usually stay at home, I go out with friends, take a walk with them; we have puri, shingara, moglai, sandwich-whatever is available and sold in the shops by the street.”*
(Adolescent boy, FGD 1)

Street foods supposedly added a dimension of joy and completeness to their socialization, bonding and attachment with fellow adolescents, which the adolescents considered to be indispensable in spite of knowing well the lack of hygiene measures in preparing and vending these foods. Besides, the adolescents suggested that these foods typically being spicy, appetizing and inexpensive fit both their taste preference and pocket money.

#### 3.3.2. Avoiding Home-Made Food Owing to Peer Pressure

The mothers elaborately expressed their worry about how the adolescents refuse to take food from home to eat at school during the break. They considered peer pressure to play a significant role in this worrying trend. It was claimed that the adolescents preferred having pocket money to buy foods from nearby vendors over carrying home-made lunch to school. This was assumed to result from the peer pressure that makes bringing food from home look like an unusual practice among classmates.


*“They feel shy. My son says, nobody brings food to school; how could I do that? They (classmates) stare at me. Initially, my son used to take noodles or roti with fried egg from home. Now he doesn’t, because his classmates don’t bring tiffin. He wants pocket money, so that he can buy those (foods sold by vendors) during break.”*
(Mother, FGD 2)

The mothers felt that persuading their children to take lunch from home to school could keep them away from buying and eating foods sold by vendors near the school or on the way to and from the school. However, they also wanted the school to adopt measures to motivate the adolescents to bring food from home, and thereby, to sort of mitigate such peer influence.

#### 3.3.3. Poor-Quality Food Environment Surrounding the School Limits Adolescents’ Food Choices

Most of the informant adolescents were school-goers. They spent greater part of weekdays in school, usually from 10 AM to 4 PM, with a lunch break around 1 PM. The paramount influence of food environment around the schools in determining what the adolescents eat day-to-day was heavily discussed. Instead of bringing lunch from home or going back to home during the break, adolescents resort to “*puri*, *shingara*, *fuchka*, *jhalmuri*”, which they could easily access near the school. Additionally, they eat street foods on their way to home after finishing school.


*“These are sold by the vendors near the school premise and we are with our friends when we are in the school, so it’s an easy option.”*
(Adolescent girl, FGD 1)

Although the foods that the adolescents can access near the school appeared to match their taste-driven food choices, these foods are of poor nutritional quality. The range of foods to choose from was quite narrow-mostly cheap, fried items or finger foods with low nutrient content. Such poor-quality food environment combined with peer influence nurtures a pattern of dietary behavior that is inclined toward street foods rather than healthier alternatives. It was not possible for the adolescents to access nutrient-rich foods, even if they opted for, in the vicinity of the schools because of the lack of availability.

### 3.4. Construction of Gender in a Rural Context Matters

The more the focus groups went deeper, the clearer the effect of gender construction in a collectivistic, rural setting on dietary and physical activity behaviors of the adolescents became. A gendered pattern of consumption, for instance, of energy drinks and some other foods, was noteworthy. Furthermore, gender socialization and gender norms appeared to palpably affect how sports played by the boys and the girls differ as well as what range of physical activities they are engaged with.

#### 3.4.1. Prevailing Gendered Pattern of Consumption and Physical Activity Behavior

The gendered pattern of consumption was most notable in case of energy drinks, which were very popular among the boys. When reflecting on this gendered pattern, the boys not only conveyed that they like how those energy drinks taste but also expressed how they perceive those drinks to confer them with masculine attributes of bodily strength and vigor. On the other hand, mothers claimed that their daughters would rather save the pocket money to buy stuffs like hairband, hair clips or bangles than spend the money on energy drinks. The boys and girls noted that boys usually prefer sweet more than the girls do, whereas girls are fond of foods and fruits which are spicy and sour: various chutneys, tamarind fruit, fuchka and jhalmuri.


*“These (energy drinks) are cold drinks, it feels great to drink these during the hot and humid days...... [I drink] to gain energy and to become strong... our friends drink these all the time; so do we. The taste is a bit different, you know!”*
(Adolescent boy, FGD 1)

Interestingly, boys linked “good” or “healthy” foods to development of a sturdy physique with muscular built which they very much desire to acquire. Girls did not share the same view, rather considered “medium built, not too fat, not too slim” as sign of a healthy adolescent physique. The girls also articulated their aspiration for “medium built” and asserted that being fat cannot be equated with good health.


*“To become healthy, I eat a lot, but in vain. [By healthy I understand] the body should be free from diseases and appear strong and robust-that’s it... ...... I am eating all types of food, yet my body... I mean, I am not gaining weight... being healthy means gaining weight and getting taller.”*
(Adolescent boy, FGD 1)


*“I think, regarding the body size, healthy girls are of medium built, not too fat, not too slim.”*
(Adolescent girl, FGD 1)

It was widely accepted and agreed upon that the responsibility of getting the household chores done falls on the girls. The girls invariably helped mothers in cooking, cleaning, washing clothes, tidying up the house and similar activities. While one mother claimed that her daughter carries out such household work “spontaneously”, others reasoned that there is undeniable difference in the nature of work that could be done by boys and by girls—girls and women stay indoors and learn household activities, thus becoming better at taking care of these chores than their male counterparts. On the other hand, the boys regularly helped their fathers in outdoor agricultural activities during the cultivation and harvesting periods, and were sparingly assigned with grocery shopping by the mothers. Girls felt that some of the works that the boys commonly take care of could not be handled by girls.


*“The (adolescent) boys do manually heavier work and those that require one to be outdoors ... Manually heavy work, such as carrying sacks of paddy or parts of irrigating machine, and cutting trees, plucking something from trees- these, we are not able to do. The boys deal with these.”*
(Adolescent girl, FGD 2)

There was striking gender-based difference in sports played by the adolescents. Contrary to the boys who enjoyed the freedom of going out to the fields and play such sports as football, cricket, and kabaddi, the girls were confined with the option of either playing local yard games that involve mild to moderate running and hopping movements, such as, *kutkut* and *bouchi*, or playing indoor tabletop games like Ludo or Carom.


*“These (bouchi, kutkut) are girls’ sports, we don’t play these... ...... the girls usually are seen playing these games, they don’t play cricket, we don’t play their games.”*
(Adolescent boy, FGD 1)

It was debated among the girls whether such gender-based difference resulted from girls having less stamina than boys for playing sports like football or because of the disapproval of the society. The girls pointed out the aversion of the community to the idea of adolescent girls pursuing outdoor physical activities.


*“I think, it’s not about the strength; my mother says, it doesn’t look good that girls are doing a lot of running and jumping out in the open; I mean, it has never been a thing appreciated in this society.”*
(Adolescent girl, FGD 2)

#### 3.4.2. Rigidity of Existing Gender Norms Impacting Physical Activity Behavior

The gender-based difference in physical activity behavior reflected to a certain extent the rigidity of existing gender norms in an agrarian, rural community. Particularly stressed was the issue of riding bicycle. Boys had no problem riding bicycle in public or to go out cycling with friends. Contrastingly, social acceptance of girls riding bicycle in public was lacking. The girls were well aware of such disfavoring social norm and were compelled to comply with this even though some of them learnt riding bicycle as children.


*“Everybody in this locality disapproves of girls riding bicycle openly. Our relatives like uncles and other older people object, they say- girls riding bicycle in public- does that look good?”*
(Adolescent girl, FGD 2)


*“In the villages, if a girl rides bicycle, she becomes a topic of gossip and harsh criticism. It’s OK if the boys ride bicycle, but if a girl does so, words and rumors start spreading!”*
(Mother, FGD 1)

Mothers discussed how riding bicycle in public could damage a girl’s reputation in that rural locality. The simple act of riding bicycle was heavily stigmatized if it was done by an adolescent girl instead of an adolescent boy. Consequently, there were families that despite being able to afford a bicycle for the daughter, refrained from doing so. The mothers acknowledged that such gender disparity has implications for physical activity required to maintain health, without addressing how this could be overcome.

## 4. Discussion

The narratives related to dietary practices were broadly constructed from two seemingly dissimilar perspectives, one that of the mothers and one of the adolescents. However, it was discernible that both perspectives converged quite often. The salient feature of the whole discourse kept relating to how mothers’ and adolescents’ perceptions contrasted with the actual dietary practices adopted by the adolescents. While mothers supported the need for nutritious diet, they were challenged by adolescents’ inclination toward street foods. On the other hand, the adolescents were quite aware of the health-compromising aspects of eating out but overlooked these owing to taste preference and peer influence. Moreover, gender emerged as an overarching influence heavily affecting the physical activity pattern through deeply-rooted cultural convention centered on gender appropriateness.

Before delving deeper, the broader context in which these mothers perform the task of cooking and of making food available for immediate consumption by the family, day in day out, merits elaborate consideration. These mothers were housewives-some with primary and secondary education, some without formal education. The traditional pattern of labor division assigns mothers with the responsibility of socially reproductive works [29], such as those concerning food, feeding, care and socialization necessary to maintain life on a daily basis. Likewise, these mothers were in charge of cooking the meals for their adolescent children, making sure they eat healthy and maintain a desired body size, deciding on food-related purchases, and most importantly, finding the balance between limited affordability and arrangement of healthy meals for the family. In doing so, these mothers consider the dietary preferences of their adolescent offspring seriously and engage in a process that sociologist Annette Lareau termed “concerted cultivation” [30]: a parenting approach where the mothers think that their children’s opinions need to be actively fostered. Hence, they take adolescents’ desires seriously, rather than strictly catering them certain foods that are regarded healthy, for instance, the vegetables, or forbidding them to eat certain foods that are regarded unhealthy, such as the street foods. The mothers paid attention to the taste-driven preferences of the adolescents without reflecting on the health impact of not having a balanced diet during adolescence. As a result, the adolescents enjoyed autonomy to a significant extent despite not being the only actor involved in decision-making around household food purchase and food allocation. Such autonomy has been linked to unhealthy eating [31]. Moreover, mothers had to deal with the lack of affordability that limits access to high-quality foods. They found it difficult to regularly feed the adolescents such foods as meat, egg and milk, which they understand to be healthy and appropriate for the younger members of the family, and therefore, resorted to saving up these for the adolescents whenever they could afford to buy these. This illustrates how these mothers negotiated with resource constraints under social pressure to epitomize caring, responsible “good mothers” [32] by prioritizing adolescents’ dietary choices.

Adolescents’ account disclosed highly taste-driven dietary preferences, as such that even their perception of lack of hygiene or of foods prepared outside of home being unhealthy did little to actually prevent them from consuming foods that they find tasty. Food taste is a crucial part of one’s identity and extensive sociological inquiry has been undertaken to understand how food choices based on liking for specific tastes are linked with social, cultural and economic contexts [33,34,35]. While Bourdieu’s pioneer work puts forward the notion of social class-based conditioning of taste [33], other studies also show that a multitude of such factors as prior taste experience and repeated exposure [36], biological conditioning [37], influence of one’s social network and social relationships [38], traditions of particular region and nation [39], and so forth. Expanding on Bourdieu’s proposition of “cultural capital”—non-material elements characterized by symbols, ideas, tastes and preferences that accumulate throughout the life and can be utilized as resources in social action—Pachucki et al. argues that tastes in food can be seen as a latent “cultural capital” shaped in large part by interpersonal mechanisms within the scope of our relationship with companions, parents and family members [38]. This argument fits what the adolescents narrated in this study well. The adolescents believed that the savory, spicy and oily foods sold by street vendors taste better than home-made foods, as they commonly enjoy the street foods while spending time with their peers and share those foods with each other, reflecting social intimacy and bonding that deploy food taste as a demarcation of peer group membership. On the other hand, one could argue that immediate availability and price play an undeniable role as Matlab adolescents are deprived of healthier alternatives which might appeal to their taste preference and the limited amount of pocket money that they receive from parents make the cheaper street foods even more attractive.

The mothers and adolescents had an immaculate understanding of the importance of hygiene and food safety, and how this relates to health status, perhaps reflecting the lessons learnt from collective memory of this community of cholera and other diarrheal diseases as well as public health efforts led by icddr,b [40,41]. Similar to an Ecuadorian study [42], the participants strongly associated healthy eating with food hygiene and consumption of home-prepared or home-grown foods, without referring to overall nutritional quality of the diet. Interestingly, participants’ perception of what constitute a healthy diet appeared to be narrower in comparison to their cognizance of how important hygiene and food safety are for health. They described such foods with high biological value as meat, milk and egg as “good foods” while the notions of diversity and balance in the diet received no attention. It is understandable that in an agricultural community where strong physical condition is of great importance due to the manual activities required during cultivation and harvesting, the foods are considered “good foods” only when perceived to be capable of boosting bodily strength and sturdy physique. In an era of “nutricentrism”—marked by evaluation of diets based exclusively on their nutritional qualities that convert individuals to “nutricentric citizens” [43] who define diets only in terms of “good and bad cholesterol”, Glycemic Index or energy requirement in kilocalories [43,44]—our participants were less aware about the multifaceted link between balanced, diversified diets and adolescents’ health. They apparently were indifferent to the caloric contribution of frequent snacking and health effects of consuming deep-fried foods. As an encouraging sign for a resource-limited setting, one mother voiced how adequate knowledge of nutrition can make them able to capitalize on locally available, free-of-cost ingredients, for example green, leafy vegetables (shaak) ubiquitous in rural areas, to diversify the diet consumed by adolescents, claiming that these leafy vegetables are in no way inferior to meat, milk and egg. On the other hand, mothers felt that such ultra-processed, empty-calorie foods [45] as “instant” noodles, mass-produced biscuits and muffins in attractive packets are a healthier alternative to the street foods that contain “germs” and “dirt”, again signaling a knowledge gap that drove a pattern of snacking that is unhealthy.

This study also uncovered multiple layers of influence exerted by peers interplaying with low-quality school and community food environment in shaping adolescents’ preferred eating behavior. Firstly, bringing food from home to school sparked the possibility of being mocked or frowned upon by classmates, reflecting strong peer norms that make healthier food choices look untrendy with an implicit risk of marginalization. Such an extent of peer influence that in this age group purportedly overpowers parental influence in establishing health-compromising dietary practices has been documented by several studies across settings [42,46,47,48]. Secondly, peer influence and low-quality food environment acted in synergy to limit adolescents’ food choices. Agreeing with previous studies [46,49,50,51,52,53], our findings distinguished the powerful influence of easy access to cheap street foods adjacent to school and on the way to and from school on adolescents’ eating habit, as they narrated on Fridays, when the school is closed, they consume fewer to no outside foods, simply because of the inaccessibility. Besides, the schools in Matlab do not operate school meal program. Matlab being a rural area, there was no clustering of fast-food restaurants near the school like that of some high-income countries [54,55], rather this space was occupied by street food vendors selling cheap, savory, excessively oily or deep-fried items with low hygienic standards that appeal to these adolescents overwhelmingly. Thirdly, sharing street foods with friends bore special meaning to the adolescents. It reinforced their social bonds and allowed them to enact their identity through eating [56], as they acknowledged that their taste preference differed from that of their parents and other senior family members, lending support to the proposition that adherence to particular dietary practices imparts a sense of belonging to a particular ethnic, geographic or age group [57].

Gender-based difference in dietary preferences is embedded in the cultural fabric of societies across the world, often through the symbolization of various foods as markers of femininity or masculinity [58]. Perhaps, the most pronounced exemplification of this is meat being considered masculine. While sociological research from the Global North captures that the idea of meat being quintessentially masculine permeates food culture [59,60,61], among our participants energy drinks were the item with the most gendered consumption, as boys unreservedly voiced its popularity among them, whereas the girls considered energy drinks to represent somewhat a “boy thing”. Of note, the brand names of the locally available energy drinks—Shark, Tiger and Speed—portrayed a macho image marked by strength, vigor and agility. A review by Visram et al. [62] identifies taste and energy-seeking attitude as drivers of energy drink consumption by young people. Accordingly, our study found that boys were fond of the taste of energy drinks and believed that drinking those would confer physical strength. This idea was propagated and augmented within their social network in a way that energy drink consumption led to attainment of a masculine sense of self, as shown by Chiou and colleagues [63]. The participants also associated rice and animal-source foods, meat, egg and milk in particular, with masculine attributes of strength and virility, suggesting that these foods, because of holding a perceived higher position than leafy vegetables or fruits, can form the core of a meal that they consider healthy for the adolescents [58]. This superseded the notion of balance or diversity in diet and mothers stressed higher cost of these foods as a significant barrier to ensuring healthy diet for adolescents.

Body image perceptions of the mothers and adolescents merit critical appraisal too. Positive socio-cultural attitude toward overweight and associated social valorization of stoutness, especially in developing countries in Africa and parts of Asia, is well-documented in literature [64,65,66,67,68]. This socio-cultural stance often appreciates being overweight as symbolic of beauty and wealth [65] as well as adds value in having big belly for men and large hips for women [66]. Nevertheless, our participants elucidated a pronounced preference for an adolescent body that is “not too fat, not too slim”. Adolescent girls went even further saying that being fat cannot represent good health. This perceptual shift away from the valorization of stoutness can be harnessed through community-based interventions to prevent overweight or obesity that has become common for countries undergoing nutrition transition [3,4]. On the contrary, the adolescent boys expressed their desire for muscular built and reportedly wanted to put on some more weight. This could signal their strong will to acquire physical attributes that they and their peers consider to be expressive of masculinity. The role of media exposure and pervasive marketing strategies in projecting such a vision of masculine body among adolescent boys has received attention in contemporary literature [1].

Remarkable internalization of gender roles was observed in relation to division of household labor as well as pattern of sports and physical activities that the adolescents engage with. Corresponding to the male breadwinner/female homemaker model [69] of the rural families in Matlab, adolescent girls took care of the traditionally feminine, monotonous household work—cleaning, tidying, helping their mothers in household chores, washing clothes—in a display of what West and Zimmerman described as “doing gender” [70]. Mothers frankly endorsed the idea that men are not meant to perform these tasks. Additionally, the rigidity of existing gender norms in a collectivistic social context was manifested through the normative engagement of adolescent girls in indoor, tabletop games or traditional games that can be played on the house yard, while the boys played football and cricket in the fields. The boys explicitly referred to those traditional games as “girls’ games”. Riding bicycle, which is a sustainable and popular way of staying physically active, was not possible for the girls to pursue in that community. The process of gender socialization, that intensifies during adolescence [71], catalyzed the internalization of community-ascribed perception that adolescent girls riding bicycle in public or playing sports like football or cricket out in the fields is inappropriate. Past research links MVPA to attenuation of cardiometabolic risk factors among adolescents [72]. However, in comparison to their male peers, not only adolescent girls are less likely to achieve the recommended level of time spent in MVPA [73,74] but also they encounter more barriers to adopting healthy physical activity behavior [18]. The strict sense of gender appropriateness of specific sporting or physical activities resulted in a lack of social support that prevented girls from embracing healthy physical activity behaviors. This gendered predicament could translate into a differentially higher health risk for these girls than the boys in upcoming adulthood.

Findings from this study have implications for adolescent health and nutrition. As lack of knowledge regarding the importance of balanced, diversified diet for adolescents was unmasked, future interventions need to address this gap. Moreover, willingness of mothers to ensure healthy diet for their adolescent children, their prominent role in feeding them and shifting body image ideals can be channeled toward better dietary practices by designing community-based, participatory nutrition interventions focusing on mother-adolescent dyads, somewhat similar to a program implemented in rural Ethiopia [75]. Broadening their perception to take advantage of locally available ingredients for improving adolescents’ diet is highly relevant for a low-income setting like Matlab. The issue in need of urgent consideration is the low-quality school and community food environment, which broadly encompasses location of food outlets and retailers, type and price of foods available, nature of foods accessible at schools, workplaces or neighborhoods, and how they are advertised, sold or vended [76]. Growing body of evidence indicates the influence of school and community food environment on dietary patterns and practices of school-goers [76,77]. Likewise, dietary practices and food choices of adolescents in the present study were shaped, to a significant extent, by their easy access to foods sold at the school gates, adjacent to schools or on their way to and back from school being reinforced by peer influence, lack of affordability and access to healthier alternatives. The school food environment in Matlab is distinct from that of urban or high-income settings reported elsewhere [54,55,78], dominated by street food vendors and small-scale retailers that typically offer very oily, deep fried, nutrient-poor street foods or empty-calorie foods like potato chips, industrially-produced snack mixes, muffins or biscuits in small, attractive packets. Consequently, fruits, salad, vegetables, dairy and whole grain products were of very limited availability. This highlights the necessity of supportive school food environment for enabling adolescents to make healthy choices. Promising avenues in this regard range from peer-led nutrition promotion program, school meal program or subsidized canteens serving healthy foods, regulation of food vending within and beyond school boundaries and so forth [79,80,81]. Like the neighboring country India, schools in Bangladesh are yet to define and adopt school food policies, whereas many Western countries have that in place [82]. The deeply entrenched, gender-based pattern of sporting and physical activity behaviors among adolescents also needs to be coherently addressed to promote well-being of these rural adolescents.

### Strengths and Limitations

This study employed focus group discussions to gain deeper insights into participants’ understanding, opinions and perceptions of the socio-cultural influences on dietary practices and physical activity behaviors of rural adolescents. Throughout the pursuit, earnest attempts were made to create a platform that allows participants to narrate and reflect on the issues as well as to agree and disagree with each other, comfortably, in a manner that resembles the natural community dynamics. This enabled exploration and comprehension of the nuances of perceptions and shared social norms. The blend of backgrounds and experiences in the research team paved the way for a broader interpretation of the findings from different cultural perspectives. The last author spent months in Matlab for research purpose, bringing back contextual awareness and acumen that ensured consistency in data analysis. Repeated discussions, formal and informal, among the authors led to a common ground of understanding of the meaning of the textual data that kept the analysis truly data-driven. The qualitative exploration in a rural context also adds novelty to this piece of research. Our approach maintained the focus on the point of view of the participants and their “words” and “stories” consciously avoiding researcher-driven, positivistic expressions [83].

However, the findings need to be interpreted in light of a number of limitations. Some of the responses elicited during the focus groups may appear to reflect socially laudable, normative articulation, as the facilitators held a superior position in social hierarchy. Careful gestures, non-hierarchical wordings of the verbal probing and prolonged engagement with the community were employed to go beyond the layer of social desirability. One can also argue that the inclusion of mothers added some degree of non-homogeneity in the overall analysis as their accounts might distort the themes emerging from adolescents’ accounts. Nevertheless, inclusion of mothers was justified by their crucial role in household activities related to food purchase, cooking and food allocation, and in parenting these adolescents. Interestingly, adolescents’ exposition corroborated mothers’ standpoint more often than anticipated. Mothers were not present during focus groups with the adolescents to encourage free and fearless discussion on dietary and gender-related themes. Furthermore, conducting focus groups with adolescents, even on relatively less sensitive dietary topics, presented intrinsic methodological challenges as they tended not to open up and fluently express themselves in front of the facilitators who they perceived to represent people who are older than them, like their parents and teachers and therefore, felt some sort of reservation in expressing themselves before them. At times, this hampered the dynamic flow of discussion that is typical of a well-functioning focus group. Future research on qualitative methodology needs to shed light on this complex methodological challenge. The transferability of specific findings from this study is left up to the readers, as findings relate to the rural, agrarian, low-income setting of Matlab. Replicating the research in other Bangladeshi settings can supplement this limitation.

## 5. Conclusions

This qualitative pursuit aimed to explore the socio-cultural influences that shape rural adolescents’ dietary practices and physical activity behaviors. The findings demonstrated adolescents’ food choices to be remarkably taste-driven inculcating a tremendous popularity of street foods and energy drinks among them. Narrow perception of what constitutes healthy diet, limited idea about the importance of balance and diversity in adolescents’ diet, suboptimal school and community food environment, peer influence, and lack of affordability act in concert to limit their access to nutrient-rich foods and health-promoting diets. Gendered pattern of sporting and physical activities pointed toward the rigidity of gender norms and community-wide approval for gendered division of household labor with clear-cut power dimension. These findings pinpoint major targets for multi-dimensional community-based interventions, which need to be tailored to the context and capable of addressing the gendered dimensions, in order to improve dietary practices and physical activity behaviors of Matlab adolescents.

## Figures and Tables

**Figure 1 nutrients-11-02916-f001:**
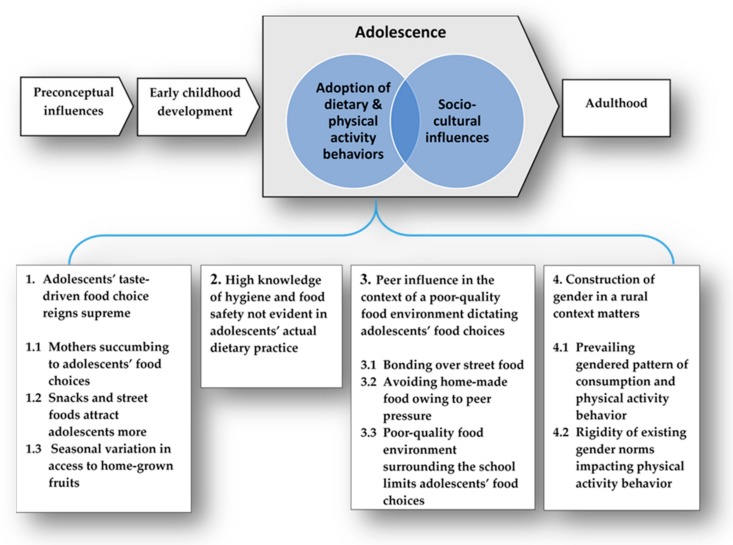
Thematic map illustrating the emergent themes from the analysis of qualitative data.

**Table 1 nutrients-11-02916-t001:** An example of the process of thematic analysis.

Data Extract	Code	Theme
*“Yes, the girls do the cooking, they can tidy the house up, they learn various household chores. The boys stay outdoors, they go and play in the fields.”* (Mother, FGD 1)	Internalized gender roles	Construction of gender in a rural context matters.
*“After finishing school, I don’t usually stay at home, I go out with friends, take a walk with them; we have puri, shingara, mughlai paratha, sandwich-whatever is available and sold in the shops by the street.”* (Adolescent boy, FGD 1)	Street food as a part of socializing with friends.	Peer influence in the context of a poor-quality food environment dictating adolescents’ food choices.

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
