# Peer review of "Sociocultural Influences on Dietary Practices and Physical Activity Behaviors of Rural Adolescents—A Qualitative Exploration"

_nutrients, 2019, doi:10.3390/nu11122916_

Round 1

Reviewer 1 Report

Thank you for the opportunity to review this manuscript which describes sociocultural influences on dietary practices and physical activity behaviours of rural adolescents

Overall

This is a very interesting manuscript that I enjoyed reading. It needs a general proof read and there are a number of formatting errors- fixing this will help make it easier to read. You also use the word etcetera a number of times which seems a bit redundant.   

Introduction

Please review first paragraph of the introduction for referencing. Although this paper has cited a lot of references there is very little in the first paragraph. Conceptual framework: I feel like you haven’t explained the rationale enough about including mothers perspectives alongside adolescents in this section and I feel it would strengthen your paper if you had more attention on it.

Method

How were the participants recruited if they did not belong to the MINImat cohort? Data analysis: I’m not that familiar with Socratic questioning mentioned in data analysis – it would be good to provide a reference here for readers to look up

Results

This section is a bit confusing though mainly because of formatting. It seems like you’ve just done a word dump of all your results and formatting is not very good. It would also be good to do a bit of a proof read ie like 353-354. Headings are not always clear or consistent to follow this section. It would read a lot better if there was consistent formatting and headings that stand out. Ie At line 165 you’ve got ‘Adolescents’ taste-driven food choices reigns supreme’ as the overall title but there is no title for Mothers succumbing to adolescents food choices (while there are titles for snacks and street foods attract adolescents more & seasonal variation in access to home-grown fruits)- it would be good to be more consistent. Line 310 it is hard to tell this is a different heading (it would be good to have some spacing so this is clearer) – also why is this in bold while the other smaller headings are not?

Discussion

Line 469: add an between play and undeniable

Author Response

Comment: This is a very interesting manuscript that I enjoyed reading. It needs a general proof read and there are a number of formatting errors- fixing this will help make it easier to read. You also use the word etcetera a number of times which seems a bit redundant.  

Response: Thank you for pointing this out. We avoided the repeated use of the word “etcetera”.

Introduction:

Comment: Please review first paragraph of the introduction for referencing. Although this paper has cited a lot of references there is very little in the first paragraph.

Response: We acknowledge this shortcoming. We have refined the the referencing accordingly (please see p. 1-2; lines 43, 45 and 50 of the manuscript)

Conceptual framework:

Comment: I feel like you haven’t explained the rationale enough about including mothers perspectives alongside adolescents in this section and I feel it would strengthen your paper if you had more attention on it.

Response: We highly appreciate this critically important input. In response, we have added the following in p. 3; lines 113-119: “Although a growing sense of autonomy marks adolescence [25], in the rural, agrarian communities of Bangladesh they live with parents. Their mothers are the ones who carry out household responsibilities of cooking as well as deciding on food-related purchases and what to cook for and feed their children. Mothers could also affect adolescents´ dietary practices through specific parenting styles and expression of normative beliefs [26]. Therefore, the role of mothers was deemed crucial considering the topic under study in that particular setting.”. Besides, importance of maternal role was further highlighted in the Discussion section (please see p. 10; lines 439-445 of the manuscript).

Method:

Comment: How were the participants recruited if they did not belong to the MINImat cohort?

Response: We have included following sentences in the Materials and Methods section ( p. 3; lines 110-113): “The Matlab HDSS keeps track of socio-demographic events through household visits by community health workers (CHWs) every two-month. The CHWs invited adolescents and their mothers for participating in the study.”

Data analysis:

Comment: I’m not that familiar with Socratic questioning mentioned in data analysis – it would be good to provide a reference here for readers to look up.

Response: Named after Socrates, this is a process of disciplined, systematic questioning for stimulating a thoughtful discussion on the ideas and assumptions under consideration. It is a dialectical approach to harnessing logical examination of one´s understanding. In our work this helped in critically appraising our explanations for the different narratives, thus complementing the reflexivity; and in reaching consensus as the team had both outsiders and insiders to the rural setting of Matlab. We have put reference as suggested (please see p. 4; line 147)

Results:

Comment: This section is a bit confusing though mainly because of formatting. It seems like you’ve just done a word dump of all your results and formatting is not very good. It would also be good to do a bit of a proof read ie like 353-354. Headings are not always clear or consistent to follow this section. It would read a lot better if there was consistent formatting and headings that stand out. Ie At line 165 you’ve got ‘Adolescents’ taste-driven food choices reigns supreme’ as the overall title but there is no title for Mothers succumbing to adolescents food choices (while there are titles for snacks and street foods attract adolescents more & seasonal variation in access to home-grown fruits)- it would be good to be more consistent. Line 310 it is hard to tell this is a different heading (it would be good to have some spacing so this is clearer) – also why is this in bold while the other smaller headings are not?

Response: As suggested, we have formatted and marked the headings and subheadings with numbers in the thematic map (Figure 1; the edited figure is attached as a separate PDF file). To be consistent, we have put numbers marking the headings and sub-headings (representing themes and sub-themes respectively) accordingly throughout the Results section. Please observe, the four themes now appear marked as 3.1, 3.2, 3.3 and 3.4 in bold font (please see p. 5, 7, and 9).

Discussion:

Line 469: add an between play and undeniable 

Response: We have added “an” in the text (p. 11; line 477):

Reviewer 2 Report

This is an important paper that has the potential to advance the field. The authors describe the sociocultural factors that influence healthy and unhealthy behaviors among adolescents, from the mother and youth perspectives. I would recommend the authors to offer greater precision in the lit. review  (when possible) local and regional studies, and contextualize the normative/developmental changes during adolescent years. Additionally, please clarify the training of those who participated in the thematic analysis (social science). The model page 5 is very helpful. I am struggling with the term "succumbing to adolescents", I feel this puts blame on to parents. To me, this theme sounds more like an opportunity for strengthening parenting skills to address adolescents' food preferences. I assume that most parents try their best in their parenting. This paper does a good job of integrating contextual factors and social support.

Author Response

Comment: This is an important paper that has the potential to advance the field. The authors describe the sociocultural factors that influence healthy and unhealthy behaviors among adolescents, from the mother and youth perspectives. I would recommend the authors to offer greater precision in the lit. review  (when possible) local and regional studies, and contextualize the normative/developmental changes during adolescent years. Additionally, please clarify the training of those who participated in the thematic analysis (social science).

Response: Because of the paucity of local and regional studies, particularly of studies from rural areas, it became quite difficult to have greater precision in the literature review. However, that also relates to the knowledge gap that our study aimed to attend to. There are quantitative studies from Bangladesh and elsewhere examining adolescents´dietary practices through food frequency questionnaire and other quantitative tools. Nevertheless, objectives and scopes of those were recognizably different. To the best of our efforts, we could not find qualitative studies shedding light on rural Bangladeshi adolescents´ dietary and physical activity behaviors. We also stated that habits and health-related behaviors established in adolescence are likely to persist well into adulthood in the Introduction section (please see p. 1; lines 41-43).

We have now briefly mentioned the diversity, in terms of background and expertise, in the research team involved in Materials and Methods section (please see p. 3-4; lines 142-142). Among those who performed the thematic analysis, the first author had training in qualitative methods as part of master´s education in global health and the second author possesses extensive experience of applying qualitative methods, including thematic analysis, at doctoral and post-doctoral levels.

Comment: The model page 5 is very helpful. I am struggling with the term "succumbing to adolescents", I feel this puts blame on to parents. To me, this theme sounds more like an opportunity for strengthening parenting skills to address adolescents' food preferences. I assume that most parents try their best in their parenting. This paper does a good job of integrating contextual factors and social support.

Response: We would like to humbly mention that the expression “succumbing to” was reported not to put blame on the mothers. We rather have a very sympathetic view regarding how these rural mothers, constrained with lack of resources, try their best to adopt the nutritional practices that they consider the best for their children. However, in the context of Matlab, they often give in to the taste-driven food choices of their children out of their desire to please and nurture them under the scarcity of resources that these mothers face; not because of apathy. We developed this view of ours in the Discussion section (please see p. 10; lines 439-445) We are of the same opinion as the reviewer that this is an important aspect for intervention.

P.S. Unfortunately, we could not find a synonym for "succumbing to" that reflects our thematic explanation.